# Tween 80 Improves the Acid-Fast Bacilli Quantification in the Magnetic Nanoparticle-Based Colorimetric Biosensing Assay (NCBA)

**DOI:** 10.3390/bios12010029

**Published:** 2022-01-07

**Authors:** Cristina Gordillo-Marroquín, Héctor J. Sánchez-Pérez, Anaximandro Gómez-Velasco, Miguel Martín, Karina Guillén-Navarro, Janeth Vázquez-Marcelín, Adriana Gómez-Bustamante, Letisia Jonapá-Gómez, Evangelyn C. Alocilja

**Affiliations:** 1Health Department, El Colegio de la Frontera Sur (ECOSUR), San Cristobal de Las Casas, Chiapas 29290, Mexico; cristina.gordillo@estudianteposgrado.ecosur.mx; 2The Network GRAAL (Grups de Recerca d’America i Africa Llatines), El Colegio de La Frontera Sur (ECOSUR), San Cristobal de Las Casas, Chiapas 29290, Mexico; anaximandro.gomez@cinvestav.mx (A.G.-V.); Miquel.Martin@uab.es (M.M.); 3Global Alliance for Rapid Diagnostics, Nano-Biosensors Lab, College of Engineering, Michigan State University, East Lansing, MI 48824, USA; 4Social Observatory of Tuberculosis Mexico, El Colegio de La Frontera Sur (ECOSUR), San Cristobal de Las Casas, Chiapas 29290, Mexico; 5Department of Human Ecology, Center for Research and Advanced Studies of the National Polytechnic Institute (Cinvestav-IPN), Merida 97310, Mexico; 6Biostatistics and Epidemiology Unit, Faculty of Medicine, Autonomous University of Barcelona, 08193 Bellaterra, Spain; 7School of Medicine, Universidad Internacional del Ecuador, Quito 170113, Ecuador; 8Sustainability Sciences Department, El Colegio de la Frontera Sur (ECOSUR), Tapachula, Chiapas 30700, Mexico; kguillen@ecosur.mx; 9Mycobacteriology Laboratory, TB Prevention and Control Program for the Highlands of Chiapas, Ministry of Health of Chiapas, San Cristobal de Las Casas, Chiapas 29290, Mexico; jandocjvm@hotmail.com; 10State Public Health Laboratory for Chiapas, Ministry of Health of Chiapas, Tuxtla Gutierrez, Chiapas 29040, Mexico; adrgomezb73@hotmail.com (A.G.-B.); lespchis@hotmail.com (L.J.-G.); 11Nano-Biosensors Laboratory, Department of Biosystems and Agricultural Engineering, Michigan State University, East Lansing, MI 48824, USA

**Keywords:** tuberculosis, mycobacterium tuberculosis, tb detection, nanotechnology, sputum smear microscopy, tween 80, paucibacillary samples, increased sensitivity, diagnostic methods

## Abstract

Despite its reduced sensitivity, sputum smear microscopy (SSM) remains the main diagnostic test for detecting tuberculosis in many parts of the world. A new diagnostic technique, the magnetic nanoparticle-based colorimetric biosensing assay (NCBA) was optimized by evaluating different concentrations of glycan-functionalized magnetic nanoparticles (GMNP) and Tween 80 to improve the acid-fast bacilli (AFB) count. Comparative analysis was performed on 225 sputum smears: 30 with SSM, 107 with NCBA at different GMNP concentrations, and 88 with NCBA-Tween 80 at various concentrations and incubation times. AFB quantification was performed by adding the total number of AFB in all fields per smear and classified according to standard guidelines (scanty, 1+, 2+ and 3+). Smears by NCBA with low GMNP concentrations (≤1.5 mg/mL) showed higher AFB quantification compared to SSM. Cell enrichment of sputum samples by combining NCBA-GMNP, incubated with Tween 80 (5%) for three minutes, improved capture efficiency and increased AFB detection up to 445% over SSM. NCBA with Tween 80 offers the opportunity to improve TB diagnostics, mainly in paucibacillary cases. As this method provides biosafety with a simple and inexpensive methodology that obtains results in a short time, it might be considered as a point-of-care TB diagnostic method in regions where resources are limited.

## 1. Introduction

Tuberculosis (TB), which is caused by *Mycobacterium tuberculosis* (*Mtb*), is a curable and preventable disease that primarily affects the lungs (pulmonary tuberculosis, PTB) and remains one of the leading causes of death from a single infectious agent. At least one third of the estimated annual cases globally are not detected and treated early [1] due to a combination of factors related to underdiagnosis and, therefore, underreporting of people with TB. Furthermore, developing countries either have limited (or non-existent) health care coverage or lack access to rapid and reliable diagnoses [1,2]. In 2020, approximately 9.9 million people worldwide became ill from TB (of which only 5.8 million were diagnosed and reported), and 1.5 million died, including 214,000 cases with TB and Human Immunodeficiency Virus (HIV) coinfection [3]. In that same year (2020), only 59% of PTB cases were bacteriologically confirmed in developing countries, while 81% were reported in high-income countries, with access to more sensitive but more expensive diagnostic tests [3]. Furthermore, it is estimated that about 2.8 million people have no access to prevent, diagnose, and care for TB [3]. The situation has worsened with the emergence of COVID-19, because human, economic, and infrastructure resources have been diverted to combat this new disease [4]. The COVID-19 pandemic has globally caused an 18% reduction in the diagnosis of new TB cases (compared to 2019) and an increase in deaths due to TB [3]. Thus, having an accessible, sensitive, and quick diagnosis is critical for reducing TB transmission, incidence, and mortality, particularly in settings with a high prevalence and limited resources.

In 2020, the Mexican National Program of Tuberculosis Control estimated 30,000 TB new cases in the country (including the HIV-seropositive population), but only 16,617 new cases were reported, with an incidence rate of 13 cases per 100,000 population. According to the current guidelines for diagnosis by sputum smear microscopy (SSM) (Table 1) [5,6,7], of all notified new TB cases, 45% were diagnosed at 1+, 25% at 2+, and 27% at 3+, evidencing the diagnostic delay [8].

In developing countries, where more than 95% of TB cases and deaths occur [9], TB diagnosis is still based on direct (unconcentrated) SSM [1,10]. Because of its relatively low cost, methodological simplicity, and minimal equipment requirements, SSM is the most widely used TB diagnostic tool and the mainstay for case detection in most low- and middle-income countries [10]. Despite its advantages, SSM has a low clinical sensitivity (20–80%) compared to culture, especially in paucibacillary individuals, i.e., samples with a low bacterial load (less than 5000–10,000 AFB/mL in the sputum sample), which represent almost half of TB cases [11,12], such as pediatric, diabetic, HIV coinfected, or malnourished populations.

Although culture is the most sensitive method and is considered the gold standard, which can detect concentrations as low as 10 AFB per mL, it can take 6–8 weeks to confirm a positive case because of the slow growth of *Mtb* [13]. In addition, culture requires highly experienced technicians to ensure an accurate examination and biosafety infrastructure.

The Xpert MTB/RIF test, and its most recent version, Xpert MTB/RIF Ultra, are much more sensitive than SSM [14]. Both are automated polymerase chain reaction (PCR)-based molecular methods that amplify a sequence of the *rpoB* gene, which is specific to members of the *Mtb complex,* within two hours [15,16]. Despite being recommended by the WHO, they are not accessible as routine diagnostic methods in countries with limited resources, due to their high costs [17].

Since SSM continues to be the main diagnostic method in countries where TB is a burden, improving its sensitivity might aid in detecting more cases [2,18,19]. In a previous study, we developed a magnetic nanoparticle-based colorimetric biosensing assay (NCBA) where glycan-functionalized magnetic nanoparticles (GMNP) were added to the homogenized sputum sample to capture, extract, and concentrate AFB before staining, at a concentration of 10 mg/mL [20]. The observable number of AFB was improved by up to 47% compared to SSM. Furthermore, we also showed that the NCBA assay could improve the classification of smears.

Two different studies have also employed NCBA assays and compared them to Xpert and culture using Löwenstein–Jensen (LJ) medium. Bhusal et al. [21] analyzed 500 sputum samples from the same number of people with respiratory symptoms using the NCBA technique for TB detection. Each sample in the study represented a pooled sputum sample collected over three days from the same patient. GMNP were added to each pooled sample at a 0.5 mg/mL concentration. Then, the sample was concentrated to a final volume of about 20 µL before performing staining and microscopy. The sensitivity of NCBA showed to be comparable to the Xpert MTB/RIF (95–100%), while SSM obtained a sensitivity of only 40% (29–52%). Additionally, the authors determined that the NCBA assay could detect concentrations as low as 10^2^ colony forming units per milliliter (CFU/mL).

In another study, Briceno et al. [22] compared NCBA to both SSM and LJ culture using 1108 sputum samples from people with a high probability of having TB. They determined that NCBA, when including 1 mg/mL of GMNP, has a sensitivity of 100.0% and a specificity of 99.7%, in agreement with cultures, while the sensitivity of SSM was 63.9%. The authors also stated that NCBA and culture could be used to detect the disease as early as the second month after onset (4.0 × 10^1^ CFU/mL), as long as the sputum sample is available [22]. NCBA would be very useful because it is fast, simple, and has a low cost.

The core principle of NCBA is cell enrichment through GMNP, due to its glycan interaction with the *Mtb* cell envelope, which is rich in complex carbohydrates, glycoproteins, and lipids [23,24]. These characteristics allow the formation of complexes between GMNP-AFB by Brownian motion, hydrodynamic force, and physicochemical and molecular interactions. Extraction of GMNP-AFB complexes is achieved using a simple magnet, where GMNP acquire superparamagnetic properties, allowing the complexes to move in the direction of the electromagnetic field and generate a highly enriched crowding effect [22]. The increase in AFB number favored by GMNP makes a substantial difference compared to unconcentrated SSM. We have shown that GMNP attach to the mycobacteria cell wall, allowing the capture, detection, and quantification of AFB in artificial and natural sputum samples [20]. GMNP have also been used to extract, isolate, and concentrate other pathogenic bacteria in food, such as *Salmonella enterica, Bacillus cereus, Staphylococcus aureus, Escherichia coli* O157:H7, *Listeria monocytogenes*, and other human pathogens [25,26,27,28].

The NCBA assay has the following advantages compared to SSM, culture or PCR-based methods: (a) homogenization and liquefaction of sputum sample eliminates non-mycobacterial cellular materials [20]; (b) it ensures the detection and concentration of *Mtb* cells in highly endemic regions; (c) it is a fast and straightforward technique, providing results in a few minutes; and (d) it has a reduced cost, since it uses the same equipment and infrastructure available for performing SSM. In spite of not being exclusive and specific to *Mtb*, the glycan–glycoprotein interaction allows the capture of AFB in symptomatic respiratory patients in TB-endemic regions because other mycobacteria are less frequently detected in sputum samples [29,30].

In our previous study [20], after cell enrichment by NCBA, we observed that GMNP-AFB complexes agglomerate. GMNP can completely cover AFB in this agglomeration, hindering microscopic observation and affecting quantification. Therefore, reducing this clumping could improve the observation and quantification of AFB. Consequently, we hypothesize that including a dispersant that disaggregates AFB-GMNP complexes would likely enhance the NCBA assay and, ultimately, the diagnosis of TB, particularly in paucibacillary cases.

Polyoxyethylene sorbitan monooleate (commonly known as Tween 80), a nonionic surfactant, has been widely used as a reductant, dispersant, and stabilizer in the synthesis of metal nanoparticles, including those of gold [31], zinc [32], silver [33,34], and iron [35]. Tween 80 is unique in its ability to enhance the steric repulsion between nanoparticles [31], demonstrating that when this surfactant is removed, nanoparticles aggregate, mainly when they are added in a solvent [36] under conditions of high ionic strength [37], as is the case in biological samples from the respiratory tract [38]. This aggregation reduces the functional surface area of nanoparticles, their mobility within the analysis matrix, and their reactivity [39].

In mycobacterial culture media, Tween 80 is added as an alternative energy source for growth [40] and to obtain homogeneous *Mtb* suspensions by reducing cell clumping [41]. It has been suggested that the interaction of Tween 80 with mycobacterial cell wall components, such as phthiocerol dimycocerosate (PDIM) [40], arabinomannan (AM), trehalose dimycolate (TDM), and mycolic acids, induces solubilization and permeabilization, altering the morphology of the bacilli [41,42,43]. Thus, we theorize that AFB quantification could be improved by adding Tween 80 in NCBA, due to its effects of dispersing clumped GMNP more stably in aqueous solutions with the presence of biological molecules (e.g., nucleic acids or proteins of different sizes, among others) [31] and the conformational changes in the *Mtb* cell envelope induced by Tween 80 could favor the disassembly of GMNP-AFB complexes, which would eventually allow their release as single bacilli without GMNP coating and improve AFB quantification in NCBA.

Considering the properties of Tween 80 and our previous observations [20], we hypothesized that combining GMNP and Tween 80 in the NCBA method would improve the visualization and quantification of AFB from human clinical samples. To prove this hypothesis, we began by optimizing the GMNP concentration in clinical sputum samples. Then, we tested the optimal GMNP concentration in different Tween 80 concentrations. Therefore, the objective of this study was to evaluate GMNP concentration in NCBA in combination with Tween 80 to improve the quantification of AFB and increase the positivity of the observed fields.

## 2. Materials and Methods

### 2.1. Chemicals and Reagents

Sodium hydroxide (NaOH), N-acetyl-L-cysteine (NALC), and Tween 80 were purchased from Sigma-Aldrich, St. Louis, MO, USA. A Ziehl Neelsen staining kit (HYCEL, SKU: 64293) was used.

The GMNP were provided by the Alocilja Research Group at Michigan State University. GMNP contain an iron (III) oxide core, also called magnetite (Fe_3_O_4_), and a glycan (chitosan) layer with a diameter of approximately 100 ± nm and an average glycan thickness of 10–50 nm. The GMNP were synthesized based on the procedure of Setterington et al. [44], with modification, by dissolving ferric chloride hexahydrate (1.08 g) and sodium acetate (2.0 g) in ethylene glycol (30 mL) for 2 h at room temperature. The solution was transferred to a Teflon-lined stainless-steel pressure vessel (Parr Instrument Company, Moline, IL, USA), sealed, and heated at 200 °C for 15 h. The pressure vessel was then cooled to room temperature. The synthesized nanoparticles were magnetically separated with a commercial magnetic separator (Promega Corporation, Madison, WI, USA), washed with 20 mL of water three times and with 20 mL of ethanol three times, and dried overnight under vacuum. Chitosan was used to biofunctionalize the magnetic nanoparticles during synthesis by adding chitosan in the ferric chloride-sodium acetate-ethylene glycol solution, essentially coating the shell of the nanoparticles with D-glucoseamine or glycosaminoglycan units.

To visualize the GMNP, a transmission electron microscope (TEM, JEOL 100CX) from the Center for Advance Microscopy, Michigan State University, was used. A TEM image of GMNP is shown in Figure 1 (adopted from Bhusal et al. [21]). The image was prepared using the negative staining method [45]. Briefly, the procedure used was the droplet technique. A drop of the magnetic nanoparticle solution was added on a copper electron microscope (EM) grating surface pretreated with 300–400 mesh. The EM grid with adsorbed particles was washed with deionized water to remove salts and macromolecules that could interfere with particle staining. Next, the EM grid was stained (e.g., with 1% uranyl acetate) to produce a thin amorphous film after drying with filter paper to reveal the final structural details of the particles. Finally, the EM grating was placed under the TEM (at 20,000x magnification) for visualization and image capture. The size of the GMNP (non-clustered nanoparticles) was measured using the Keyence VK-X150 laser scanning confocal microscope system measurement software (Nano-Biosensors Lab., Michigan State University, East Lansing, MI, USA).

### 2.2. Clinical Samples

From 29 January 2019, to 30 April 2020, 46 positive sputum samples were collected from the Mycobacteria Laboratory at the Center for Comprehensive Care of Prevalent Diseases (CAIEP) in San Cristóbal de Las Casas, Chiapas, Mexican Ministry of Health, which is the institution responsible for the diagnosis and surveillance of TB in the country. Negative samples to SSM, as well as positives from patients who were receiving anti-TB treatment or having a volume of sputum less than 1 mL, were excluded.

The 46 positive samples were from 32 patients, with 20 belonging to women (43.5%) and 26 belonging to men (56.5%). The mean age was 43 years with a standard deviation (SD) of 16 years. Samples were collected in sterilized screw cap bottles. In all cases, patients’ personal information was discarded (except for sex and age) and underwent recoding for this study.

Unconcentrated SSM was conducted immediately upon arrival at the laboratory with an average time of nine hours from sample collection to processing, and the samples were categorized based on the number of AFB observed, according to the WHO guidelines (Table 1) [5,6]. The remaining samples were kept refrigerated until the NCBA was performed but for no more than 72 h.

Due to the difficulty of obtaining single samples with high volumes that would allow the evaluation of the GMNP and Tween 80 concentrations of interest in NCBA, the 46 samples were divided into two equal parts: 23 samples were processed independently. The remaining 23 samples were randomly divided into 7 pools. The 7 pools were grouped as follows: pool 1, with 7 samples; pool 2, with 4 samples; pool 3, with 3 samples; pool 4, with 3 samples; pool 5, with 2 samples; pool 6, with 2 samples; pool 7, with 2 samples. The mean volume of the 23 samples, evaluated independently, was 3.71 mL (SD = 2.78), and the mean volume of 23, grouped into seven pools, was 15.85 mL (SD = 3.52).

Using aliquots of all individual and pooled samples, a total of 225 smears were prepared and examined, as shown in Figure 2.

### 2.3. Processing of Sputum Samples

Firstly, clinical samples arrived at CAIEP and were processed by SSM. Secondly, the samples were homogenized, and then we included GMNP in the samples at different concentrations (to detect and capture AFB from sputum samples). Finally distinct concentrations of Tween 80 were included as a dispersant of GMNP-AFB complexes to evaluate its effect from high to low GMNP concentrations. Each step is described sequentially for both the method and results sections.

For AFB quantification, all smears were examined with a brightfield microscope (Carl Zeiss Primo Star model) under an oil-immersed objective (100x). The bacillary load on the stained slides was counted as the total number of AFB observed per field, in 100 fields on each slide, for SSM, NCBA, and Tween 80.

#### 2.3.1. Sputum Smear Microscopy (SSM)

SSM with the Ziehl-Neelsen (ZN) staining technique was performed with an aliquot of approximately 20 µL of each sample and each pool, according to the current guidelines for low-income countries [5]. After staining, the slides were air-dried and examined by bright-field microscopy for the presence of AFB that had stained red.

#### 2.3.2. Magnetic Nanoparticle-Based Colorimetric Biosensing Assay (NCBA)

Once the SSM was completed, the rest of the samples were homogenized with 0.4% NaOH and 4% NALC at a 1:1 proportion, which are the optimized concentrations obtained from our previous study [20]. After homogenization, 1 mL aliquots of the samples were transferred into microtubes that contained GMNP at final concentrations of 0.25, 0.5, 0.75, 1.0, 1.25, and 1.5 mg/mL (low concentrations) and 1.75, 2.0, 2.25, 2.5, and 10.0 mg/mL (high concentrations).

Aliquots were mixed by inversion followed by incubation for 10 min at room temperature (~18 °C). Microtubes were then placed in a 3D-printed magnetic rack to separate the GMNP-AFB complex, and the supernatant was discarded. A smear was prepared by transferring 20 µL of the concentrated sputum to a slide that was then stained using the ZN technique [5].

#### 2.3.3. Evaluation of the Tween 80 Effect in the Dispersion of GMNP-AFB Complexes

To analyze the effect of Tween 80 on the dissolution of the GMNP-AFB complexes formed during NCBA, two procedures were developed: (a) For high concentrations of GMNP and Tween 80, initially, 24 aliquots of 20 µL (two from each of the 12 independent samples) were processed with a high GMNP concentration (10.0 mg/mL), followed by the NCBA method. Subsequently, these 24 aliquots were separated into 2 groups: 12 were incubated for 3 min at room temperature with Tween 80 at 10% *v*/*v* concentration and 12 at 20% [46]. (b) Due to the limited results obtained in the previous procedure, we further evaluated GMNP concentrations (low-high) and Tween 80 at different concentrations (5, 10, 15, and 20% *v*/*v*) and two incubation times (3 and 10 min) experimentally determined by the authors to favor the modification of the mycobacterial cell wall, without exceeding the interaction time of the aliquots with the homogenizing solution and Tween 80. We analyzed 32 aliquots of approximately 1 mL obtained from 4 pools: 16 were concentrated with GMNP at 1.5 (the optimal low concentration analyzed, *p* < 0.05), and 16 were concentrated with 2.5 mg/mL (the highest concentration analyzed before the 10 m/mL, which showed the worst results).

For AFB quantification, all the aliquots (20 µL) were smeared onto a glass slide and stained with the ZN [5].

### 2.4. AFB Quantification of Acid-Fast Bacilli on Smears

AFB quantification of SSM used the standard WHO guidelines, as shown in Table 1 [5,6,7].

### 2.5. Statistical Analysis

To analyze trends in the number of AFB quantified in SSM, NCBA (0.25–10 mg/mL GMNP concentrations), and Tween 80 (5–20% concentrations and 3–10 min incubation times), descriptive statistical analysis was conducted in three stages: univariate, bivariate, and multivariate. The analyses were expressed by means, SD, standard errors of the mean (SEM), and by medians and interquartile ranges (IQR). The latter is due to the non-parametric behavior of the data, thus allowing to adjust for the dispersion of the results by discarding extreme values far from the median.

To determine whether NCBA at different GMNP concentrations had a favorable effect on AFB quantification and field classification in smears, with respect to SSM, McNemar’s test was conducted [47]. Pairwise comparisons between SSM, NCBA [10 mg/mL], and Tween 80 were performed by Student’s *t*-tests. To compare the effects of Tween 80 (at different concentrations and incubation times, 3 and 10 min) and the NCBA assay (with different GMNP concentrations), a multifactorial ANOVA was performed on the number of AFB quantified, followed by Bonferroni’s or Tukey’s HSD post-hoc tests [48]. When the ANOVA homogeneity of variance test showed non-normal distributions for various groups, Friedman’s test was performed, followed by Wilcoxon signed-rank test for between-group comparisons [49]. Finally, to compare the number of countable bacilli in smears evaluated in SSM, NCBA with GMNP (1.25 and 2.5 mg/mL), and with Tween 80 (5, 10, 15 and 20%), the percentage variation was calculated as in Equation (1), where V1 represents the total number of AFB counted in Tween 80 and V2 represents the total number of AFB counted from SSM or NCBA (depending on the comparison being performed):(1)$$Percentage\;variation\;\left(\%\right)=\frac{v1-v2}{v2}*100$$

All analyses were performed using SPSS version 21 (IBM Corp) (Stata Corp LP, College Station, TX, USA) at a 95% confidence interval (CI95%, α = 0.05), and the graphs were made in Excel (Microsoft Office 365).

From each smear of all samples processed by SSM and NCBA-GMNP (regardless of concentration), AFB were counted per field. Based on this count, AFB groups were categorized as: negative (non-AFB), scanty (1–9 AFB), 1+ (10–99 AFB), 2+ (100–9,999 AFB), and 3+ (>10,000 AFB) [5,6]. For smears analyzed with Tween 80 (concentration and incubation time periods), the number of AFB per field was quantified and subsequently grouped according to the GMNP concentrations that were included for the formation of GMNP-AFB complexes.

## 3. Results

### 3.1. Analysis of AFB Detection in SSM and NCBA

In SSM, the median for the number of AFB per field analyzed was 9 (IQR, 1–44.8), which was lower compared to NCBA at low GMNP concentrations (0.25, 0.5, 0.75, 1.0, 1.25, and 1.5 mg/mL). On the other hand, at high GMNP concentrations (1.75, 2.0, 2.25, 2.5, and 10.0 mg/mL), the median number of AFB in NCBA was lower than in SSM (Table 2).

The greatest variability in AFB detection was shown in the proportion of observed fields that were determined to be negative by SSM but were determined to be positive with NCBA, varying between 13.9–65.3%. The AFB count obtained as scant bacilli levels in SSM, increased between 6.5–38.2% at 1+ and 2+ with NCBA, and the SSM categories at 1+, increased to 2+ and 3+ by 1.8–27.1% (Table 3).

### 3.2. Low GMPN Concentrations Improve AFB Quantification

We found significant differences among the GMNP concentrations evaluated in the NCBA assay. Low GMNP concentrations (≤1.5 mg/mL) increased the proportion of the number of AFB per field by more than 25%, as can be seen in Table 3 At a concentration of 0.25 mg/mL, the improvement was 51.2% (of 127 fields read as negative with SSM, 65 were read as positive with NCBA); at 1.0 mg/mL, the improvement was 65.3%; and with 1.25 mg/mL, the improvement was 50.5%. Low GMNP concentrations also improved the classification of positivity fields. Fields with scanty bacilli in SSM became positive at 1+ and 2+ (16.0–38.2%) and the proportion of fields classified as positive at 1+ in SSM that became positive at 2+ and 3+ with NCBA was 11.2–27.1% (Table 3). On the other hand, with higher GMNP (≥1.75 mg/mL) or bacilli concentrations, the capture and quantification of AFB efficiency were reduced (Table 3). Using GMNP at 10 mg/mL, there was no improvement in the quantification of AFB, even after attempting to disaggregate the GMNP-AFB complexes with high concentrations of Tween 80 (10 and 20%) (see Section 3.3).

Regarding the improvement of the number of AFB per field, we also observed an increase in the average AFB count in the NCBA at low GMNP concentrations compared to SSM. After the initial classification of SSM smears as negative, scanty, 1+ or, 2+ and 3+, in the NCBA we observed 2–7, 5–16, 56–79, and 188–2030 more AFB per field, respectively. With high GMNP concentrations, the average AFB number compared to SSM, using the same categories, was reduced, as shown in Table 4.

In all cases, NCBA increased the percentage of the classification of positive fields obtained in SSM, mainly in those where the AFB loads were insufficient to classify them as positive. Figure 3 shows micrographs of aliquots of samples in which AFB were visualized when analyzed by SSM and NCBA with different GMNP concentrations. It is noteworthy to mention that when NCBA was performed, a higher number and agglomeration of AFB was observed compared to SSM. Furthermore, the visualization and quantification of AFB were favored when low GMNP (≤1.5 mg/mL) concentrations were added.

### 3.3. Tween 80 at Low Concentration Improves the Quantification of AFB in NCBA

Initially, we evaluated high concentrations of Tween 80 (10 and 20%), and we observed that these concentrations did not sufficiently disaggregate GMNP (at 10 mg/mL)-AFB complexes; hence, AFB observation and quantification were not improved.

Figure 4 shows the average number of AFB quantified after including Tween 80 at 5–20% (incubating for 3 and 10 min) in NCBA with GMNP at 1.25 and 2.5 mg/mL. The best result was obtained with the combination of NCBA-GMNP at 1.25 mg/mL, Tween 80 at 5%, and an incubation time of 3 min, which provided the highest mean value of 24 AFB (SEM ± 1.2). By changing the Tween 80 concentration in this combination at 10–20%, the average count decreases inversely (the higher the concentration, the lower the count) (Figure 4).

On the other hand, with GMNP at 2.5 mg/mL, the combination of Tween 80 concentration from 10–20% and 10 min of incubation time, the number of AFB quantified was lower. In addition, the mean values of AFB counted were not significantly better compared to NCBA (black line in Figure 4) and SSM (light blue dotted line in Figure 4).

The use of Tween 80 as a dispersant of GMNP-AFB complexes in NCBA increased the number of AFB by 3–199% compared to NCBA (without using Tween 80) and 29–445% compared with SSM (Table 5). The combination of NCBA-GMNP (1.25 mg/mL), Tween 80 (5%), and incubation for 3 min allowed the highest quantification of AFB, with an improvement of 199% compared with NCBA and 445% with SSM. When Tween 80 at 10, 15, and 20% were included, it only improved the number of AFB quantified with NCBA by 138, 63, and 41% and with SSM, by 335, 197, and 158%, respectively.

In contrast, using the combination of GMNP (2.5 mg/mL) and Tween 80 (5–20%), the percentage increase of bacilli was lower in NCBA by 7–76% and in SSM by 29–113%.

Furthermore, independent of the concentration of Tween 80, with an incubation time of 10 min, the percentage of AFB quantification decreased compared to a low incubation period (Table 5).

## 4. Discussion

Despite its reduced sensitivity, in many developing countries, SSM remains the main diagnostic test for detecting TB, as well as for assessing anti-tuberculosis treatment progress [10].

Our previous study demonstrated that, during the NCBA-GMNP assay, complexes are formed with AFB, which in turn induces cell enrichment of positive sputum samples [20] and improves the sensitivity of SSM by at least 35%. Likewise, the work by Bhusal et al. [21] and Briceno et al. [22] using the NCBA technique yielded high sensitivity and specificity compared to Xpert MTB/RIF and culture, respectively, as reference standards. Such improvement is explained by the fact that during NCBA-GMNP assay, there is the formation of complexes with AFB that, when extracted and concentrated from the sample matrix with a common magnet, increase the proportion of AFB observed in smears [20]. However, the GMNP-AFB complexes tend to agglomerate and AFB are covered by GMNP, hindering microscopic observation and quantification.

Surfactants, such as Tween 80, are a type of organic compounds with polar (hydrophilic) and non-polar (hydrophobic) groups that disintegrate nanoparticles through the repulsive forces of static electricity, steric hindrance, and Van der Waals force by their surface adsorption [50,51]. By changing the electric potential, surfactants increase the surface charge and repulsion between particles to reduce agglomeration [51]. The dispersant properties of Tween 80 have also been used in the disaggregation of mycobacteria in culture media [41], and Tween 80 has shown that it can affect bacterial adhesion by inhibiting the formation of biofilms, as occurs in *Pseudomonas aeruginosa* [52], *Salmonella enterica* [53], and *Staphylococcus aureus* [54], among others.

In this study, we used positive sputum samples to improve the NCBA assay by evaluating GMNP and Tween 80 at different concentrations to improve AFB detection and quantification compared with conventional SSM. According to current guidelines [5,6], we were also able to reclassify the observed fields initially detected as negatives or scant by SSM to either 1+, 2+, or 3+ by NCBA. The fact that AFB counts with SSM were lower than those obtained with NCBA with low GMNP concentrations (≤1.5 mg/mL) may be due to higher AFB capture efficiency (Table 2 and Table 3). Despite the difference in research design and the number of samples analyzed, our results are in agreement with the study by Briceno et al. [22] in which the authors used a GMNP concentration of 1.0 mg/mL to analyze 1108 samples with SSM, without homogenization, of which 122 were positive, while 194 were positive with NCBA.

We hypothesize that low GMNP concentrations do not saturate the matrix sample, allowing AFB their easy mobility to be captured and concentrated. In a study by Phenrat el al. [55], they evaluated the effect of iron nanoparticle concentration, size distribution (polydispersity), and magnetic attraction forces on agglomeration and transport in a water-saturated sand column, determining that low concentrations of nanoparticles (0.03 mg/mL) could easily move in the sand column regardless of size or magnetic attraction forces between nanoparticles. Although we used different analytical conditions (nanoparticle design and analytical samples) and in our study we exceeded the “low” concentration evaluated by Phenrat el al. [55], we obtained similar results regarding the efficient use of low GMNP concentrations with respect to those that are higher. In this condition (low GMNP concentration and facilitated mobility), the diversity of *Mtb* cell wall receptors can form complexes with the GMNP by Brownian motion, hydrodynamic force, and physicochemical and molecular interactions [22]. These factors provide better capture efficiency and prevent further agglomeration of GMNP in the smears, improving visualization and microscopic quantification of AFB (Figure 3).

Regardless of the GMNP concentration used during the NCBA assay, we have observed that GMNP-AFB complexes can cluster in such a way that not all AFB are visible during microscopic analysis. While some fields are highly positive, others may have scanty bacilli, as shown by the proportion of positive fields (Table 3). However, this was significantly improved when Tween 80 was added.

The addition of Tween 80 during NCBA-GMNP assay provided better observation and quantification of AFB in microscopy analysis (Figure 4 and Table 5). The combination of GMNP (1.25 mg/mL), Tween 80 (5%), and incubation for 3 min allowed the percentage of AFB quantification to increase by 445% and 199% compared to SSM and NCBA, respectively (Table 5).

The increased visualization and quantification of AFB is facilitated by two factors. (i) The low GMNP concentration does not saturate the sample matrix, as explained above and (ii) the ability of Tween 80 to increase the separation between GMNP [31] and change mycobacterial cell wall morphology [56]. With the addition of Tween 80 at 0.5% or higher concentrations during planktonic growth of *Mycobacterium avium* complex (MAC), cells become morphologically smoother and elongated, due to the loss of its various constituents, including arabinose, glucose, mannose, arabinomannan, and phthiocerol dimycocerosate (PDIM) [56,57,58].

The remodeling of the mycobacterial cell surface caused by Tween 80 presents a double advantage in NCBA. In addition to the mentioned disaggregation of GMNP-AFB complexes it also functions as an additive that could improve biosafety during laboratory work. This is because Tween 80 influences the loss of virulence factors, such as PDIM [59] and TDM [60], which are key players during the initial pathogen-host interaction by their association with MARCO, TLR2 [61], and Mincle [62] human receptors expressed on macrophages and dendritic cells, which reduces their infective property, as observed in animal models [63,64].

The usefulness of Tween 80 at low GMNP concentrations may also be favored in TB paucibacillary samples. In this study, we determined that when the number of AFB is greater than 10,000 per field, the quantification and proportion of AFB is reduced (Table 3 and Table 4). This finding is similar to our previous study in which GMNP capture efficiency (at 5.0 mg/mL) was analyzed using artificial sputum and different concentrations of *Mycobacterium smegmatis* (*Msm*), determining that this efficiency decreases linearly with increasing bacterial concentration [20].

On the other hand, the evaluation of high GMNP concentrations (≥1.75 mg/mL) continued causing a greater number of GMNP-AFB complex clusters, which reduced the visualization and quantification of AFB (Figure 3 and Table 2 and Table 4), as well as the proportion of positive fields enhanced with NCBA compared to SSM (Table 3). This effect was also observed by Phenrat et al. [55], where high concentrations of nanoparticles (up to 6 mg/mL) substantially increased agglomeration, due to the binding between primary particles, as between aggregates of them, a process that also becomes more sensitive to particle size distribution. In this sense, the addition of Tween 80 (5–20%) and incubation times did not improve the visualization and quantification of AFB. This possibly occurred due to significant Van der Waals interactions that increased the closer that the nanoparticles were to each other. At the same time, Brownian motion provided persistent collisions between nanoparticles, leading to agglomeration [34], as well as to the fact that the observation of AFB depends on the saturation of receptors offered by *Mtb* cell wall glycolipids or glycoproteins to interact with GMNP [20] and on the formation of GMNP-AFB complexes per field observed under the microscope [21]. Since the mycobacterial surface is saturated at high GMNP concentrations, it is likely that the AFB are entirely covered by them and cannot be quantified, as observed in previous studies [20,22]. Supporting this observation, complexes of *M. smegmatis* (*Msm*) and GMNP have been observed during transmission electron microscopy (TEM) analysis, where GMNP binds on the bacterial surface [19]. The study of Briceno et al. [22] suggests that, in the presence of a magnet, GMNP-AFB complexes acquire superparamagnetic properties, due to GMNP, allowing them to move in the direction of the electromagnetic field and generate a highly enriched crowding effect. This effect might explain that at high GMNP concentrations, the addition of Tween 80 is still insufficient to disaggregate GMNP-AFB complexes, as observed in this study. In support of this observation, an early study showed that with Tween 80 at higher concentrations, the bacillary agglutination during planktonic growth of *Mycobacterium avium* complex is gradually reduced by the loss of GPLs [58].

However, in our previous study, where high GMNP concentrations (10 and 20 mg/mL) were analyzed, the gain in AFB counts was 47% in NCBA, with respect to SSM with a concentration of 10 mg/mL [20]. The differences observed between these two studies may be due to the following factors: (a) an unequal distribution of the mycobacterial population in each sample and smear, which determines the number of AFB per field; (b) the nature and quality of the sputum (which influences in the quantity of AFB, the viscosity of the sample, and the homogenization and magnetic separation times during NCBA assay); (c) the homogenization time (the longer the time, the greater the mucopurulence, viscosity, and cellular load) and magnetic separation (the shorter the time, the lower the bacillary load); (d) the viscosity of the samples (the higher the viscosity, the greater the reduction of GMNP mobility in the sample matrix and the formation of GMNP-AFB complexes) [20]; (e) the differences in the size of the iron core (100 ± 58 nm) and the thickness of the glycan layer (10–50 nm) of GMNP (synthesis conditions that may be sources of variation that should be taken into account in future studies); (f) the concentration and stability of the GMNP stock; and (g) the variation in performing smears (size, thickness, uniformity, and reading) by microscopists, which can contribute from 49.1 to 72.7% in the quality of AFB detection [65].

### Novelty of the Study and Its Main Application

The clinical and microbiological classification of TB infection is currently considered as a dynamic continuum from *Mtb* infection to the various degrees of disease severity. When in an individual TB infection progresses to a subclinical form, in which there are no typical symptoms of TB, the SSM is often negative, because they are usually paucibacillary, but cultures can be positive [66]. Numerous studies have demonstrated that asymptomatic individuals are also a potential source of infection. Rasool et al. [16] have documented the transmission of TB in cases that are not SSM-positive but are positive to molecular tests and cultures. In their study, those authors described that, after testing 168 patients initially diagnosed as SSM-negative, 48 (28.6%) were classified as *Mtb*-positive with GeneXpert MTB/RIF and 58 (34.52%) with culture [16].

For paucibacillary cases, such as pediatric TB, or people with comorbidities (HIV, diabetes, or malnutrition), the SSM result is usually negative due to the low number of bacilli in the samples [66]. In these situations, having a sensitive, rapid, and cost-effective diagnostic test that detects cases with reduced bacilli populations is crucial to reduce TB transmission, incidence, and mortality rates [2,18,19].

According to our results, using 1.25 mg/mL GMNP concentrations incubated with Tween 80 at 5% for 3 min allows optimal capture efficiency to detect a higher number of AFB in fields where they were scanty in SSM. Thus, the NCBA-GMNP assay represents an advantage with respect to SSM (due to its low sensitivity) in the detection of subclinical or paucibacillary cases, with the same performance as of standard culture [22] and molecular test (GeneXpert MTB/RIF) [21].

In addition, NCBA shows numerous advantages compared to SSM: (a) the ability of NCBA to form GMNP-AFB complexes and concentrate AFB [20], through the glycan-glycoprotein interaction between the GMNP surface and the mycobacterial cell wall [21]; (b) after cell enrichment with GMNP, disaggregation of complexes with Tween 80 improves the microscopic quantification of AFB; (c) a larger volume of sample analyzed (by concentrating it with GMNP) in the same proportion of examined area may facilitate the ability to obtain higher AFB counts [22]; (d) the AFB detected and captured by GMNP, followed by ZN staining of concentrated aliquots, allows easy identification by colorimetry, which may increase the possibility to identify bacilli from paucibacillary cases [21]; (e) it is a fast method, since the complete processing can be obtained in 30–40 min from sample acquisition to final diagnostic result; (f) it is a direct and quantitative detection technique where bacilli can be viewed and counted through a microscope; (g) it is a low-cost method: one test, including materials, labor and infrastructure overhead, costs approximately USD 3.20 for NCBA and USD 4.20 with Tween 80, compared to other methods, such as culture (USD 16.50) [67], and molecular tests, such as GeneXpert (USD 12.90) [68]; (h) it does not require complex infrastructure, allowing its applicability in settings where TB is a burden, but resources are limited; and (i) offers bio-safety, since the NCBA-GMNP assay does not require centrifugation to concentrate AFB, thus reducing the generation of infectious aerosols, and the addition of Tween 80 may also reduce infectivity, due to the loss of virulence factors from the mycobacterial cell wall [60,61,64].

The limitations of the study are the following: (1) we could not analyze more positive samples due to the deviation of human, material, and infrastructure resources to combat the COVID-19 pandemic. Health staff did not carry out active and passive TB case-finding, and the access to the biosafety laboratory was limited [69]. (2) The volume of the sputum samples obtained in this study restricted the analysis of all GMNP and Tween 80 concentrations. The volume of the samples varied between 3.71 mL (SD = 2.78) for samples analyzed independently and 15.85 mL (SD = 3.52) for the pools. Despite the higher volume in the latter, we could not perform all the experiments proposed; (3) the variable quality of the sputum samples, either due to their conditions of origin or storage. In our case, even though no sample was processed after 72 h in refrigeration, which is the recommended time in the laboratory standards, we had to store some of them more than 24 h in refrigeration, which could eventually reduce the viability of the bacilli and change their morphology [5] and, finally, affect the NCBA.

## 5. Conclusions

Cell enrichment of paucibacillary sputum samples by combining NCBA-GMNP (1.25 mg/mL) incubated with Tween 80 (5%) for 3 min improves capture efficiency and increases AFB detection up to 445% over conventional SSM. NCBA with Tween 80 offers the opportunity to improve diagnostic accuracy and biosafety with a simple and inexpensive methodology that obtains results in a short time, so it might be considered as a point-of-care TB diagnostic method in regions where resources are limited. NCBA can become an essential tool in detecting paucibacillary cases, such as pediatric cases and TB patients with comorbidities (diabetes, HIV, and malnutrition, among others), which usually give negative results in SSM. Further studies will focus on analytical and diagnostic validation to determine the contribution of NCBA in TB case detection.

## Figures and Tables

**Figure 1 biosensors-12-00029-f001:**
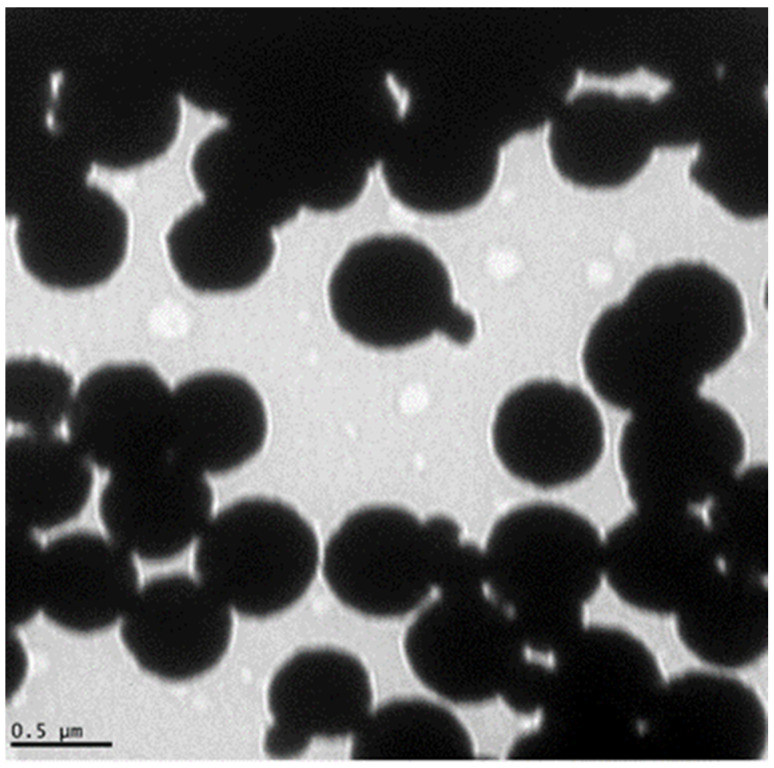
Transmission electron microscope (TEM) image of the glycan-coated magnetic nanoparticle clusters, with several iron oxides enclosed in the glycan polymer. Some nanoparticles are protruding from the cluster. Adopted from Bhusal et al. [21].

**Figure 2 biosensors-12-00029-f002:**
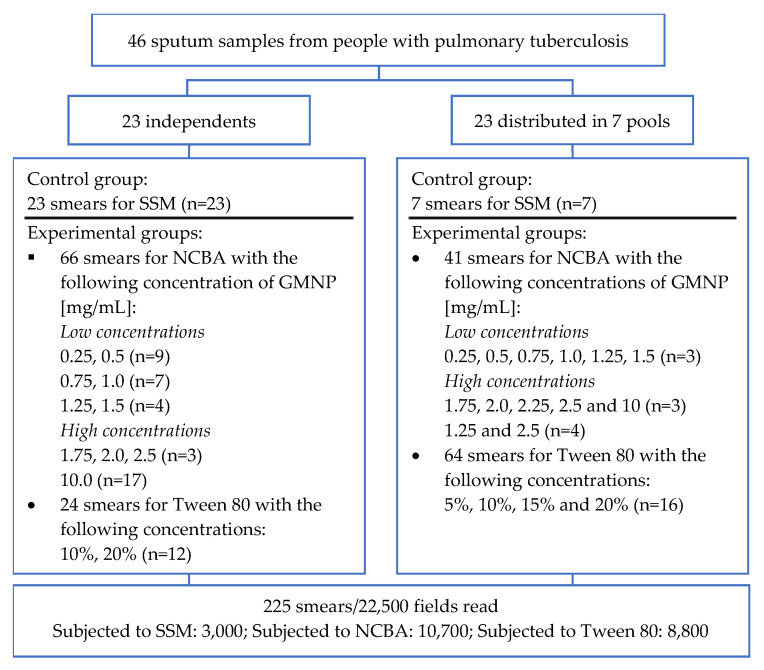
Flow diagram of the clinical samples analyzed and the experimental tests performed. GMNP: glycan-functionalized magnetic Nanoparticles; NCBA: magnetic nanoparticle-based colorimetric biosensing assay; SSM: sputum smear microscopy. The number of aliquots used for each of the smears analyzed by SSM and different NCBA or Tween 80 concentrations are shown in parentheses.

**Figure 3 biosensors-12-00029-f003:**
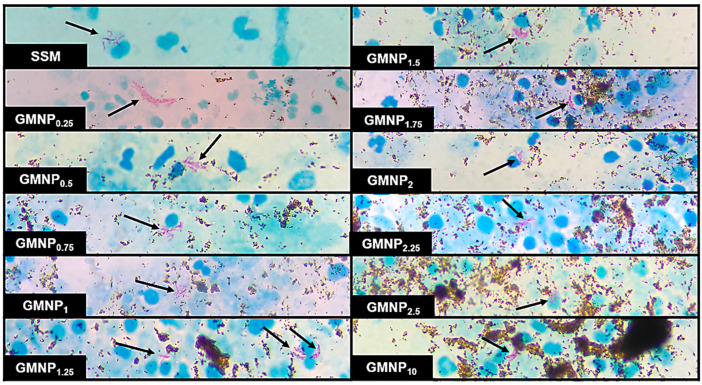
Micrographs of a positive TB sample analyzed by SSM and NCBA showing clustered purple-magenta bacilli (indicated by arrows) and GMNP in brown. Subscripts indicate the concentration of GMNP used.

**Figure 4 biosensors-12-00029-f004:**
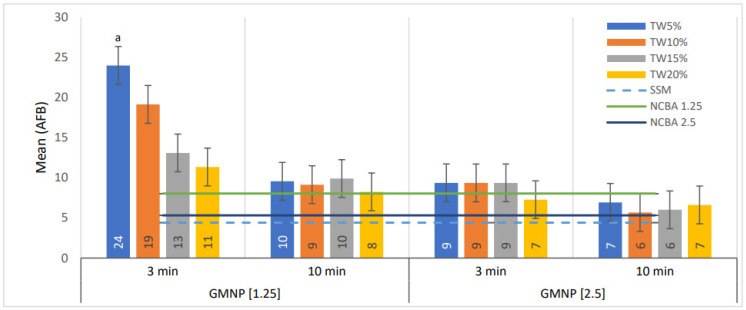
Comparison of AFB means in smears, evaluated at GMNP concentrations of 1.5 and 2.5 mg/mL, incubated with Tween 80 at 5–20% for 3 and 10 min. Statistical comparisons were made between data sets based on GMNP, Tween 80 concentrations, and incubation time by ANOVA. The numbers in the bars represent the mean values. The letter a indicates statistically significant differences for the rest of the categories analyzed (*p* < 0.05). The error bars represent the 95% confidence intervals. *n* = 4 pools.

**Table 1 biosensors-12-00029-t001:** AFB quantification guide and estimated AFB concentration per milliliter.

No. AFB Observed	AFB Quantification	Estimated AFB Concentration/mL
0 in 100 or more fields	Negative	0
1–9 AFB/100 fields	Scanty	30,000 (3 × 10^4^)
10–99 AFB/100 fields	1+	50,000 (5 × 10^4^)
1–10 AFB/field in at least 50 fields	2+	100,000 (1 × 10^5^)
>10 AFB/field in at least 20 fields	3+	500,000 (5 × 10^5^)

Source: International Union against Tuberculosis and Lung Disease (IUATLD)/World Health Organization scale (WHO) [5,6,7].

**Table 2 biosensors-12-00029-t002:** Descriptive values and mean differences of the number of AFB after performing the NCBA and SSM technique in aliquots of sputum samples from people with pulmonary tuberculosis.

Method	GMNP (mg/mL)	No. Smears	No. Fields Read ^a^	Median (AFB)	IQR
SSM	-	30	3000	9	1−44.8
NCBA	0.25	12	1200	22	14−78
	0.5	12	1200	28	4−112
	0.75	10	1000	40	2−121.5
	1.0	10	1000	48	14−142
	1.25	11	1100	10	2-80
	1.5	7	700	34	4−99.5
	1.75	6	600	6	0−52
	2.0	6	600	4	0−38
	2.25	3	300	0	0−6.75
	2.5	10	1000	4	0−51
	10.0	10	2000	0	0−42
	Total	137	13,700	-	-

Abbreviations: IQR: interquartile range. ^a^ Sum of the 100 fields evaluated by smears obtained from aliquots of independent samples (*n* = 12) and pools (*n =* 7).

**Table 3 biosensors-12-00029-t003:** Proportion of SSM smear fields (*n* = 3000) improved by NCBA in sputum samples from people with pulmonary tuberculosis.

GMNP (mg/mL)	No. Fields/ Smears	Fields SSM Negative Improved with NCBA to S+, 1+, 2+ and 3+		Fields SSM S+ Improved with NCBA to 1+ and 2+		Fields SSM 1+ Improved with NCBA to 2+ and 3+	
		Fields *	%	Fields *	%	Fields *	%
0.25	1200	65/127	51.2	117/306	38.2	89/457	19.5
0.5	1200	39/127	30.7	98/306	32	91/457	20
0.75	1000	32/126	25.4	39/250	16	67/314	21.3
1.0	1000	47/72	65.3	76/224	34	44/394	11.2
1.25	1100	102/202	50.5	111/437	25.4	83/306	27.1
1.5	700	33/72	45.8	56/224	25	62/249	25
1.75	600	31/72	43.1	30/206	14.6	3/167	1.8
2.0	600	19/72	26.4	30/206	14.6	9/167	5.4
2.25	300	15/67	22.4	8/124	6.5	2/56	3.6
2.5	1000	89/202	44.1	67/419	16	16/224	7.1
10.0	2000	76/546	13.9	41/566	7.2	64/604	10.6
Total, ≤1.5	6200	318/726	43.8	497/1747	28.4	436/2177	20.0
Total, 0.25−10	10,700	548/1685	32.5	673/3268	20.6	530/3395	15.6

Abbreviations: S+: positive with scanty bacilli. Classification based on IUATLD scale [5,6]: Negative (0 AFB); Scanty (1–9 AFB); 1+ (10–99 AFB); 2+ (100–9999 AFB); 3+ (>10,000 AFB). * The relation is given for fields observed in NCBA/SSM from all independent samples (*n* = 23) and pools (*n* = 7). For the classification of smears with respect to SSM, McNemar’s test was performed. All values were significant, with *p* < 0.05.

**Table 4 biosensors-12-00029-t004:** Differences in the number of AFB obtained per field in SSM and NCBA at different GMNP concentrations in sputum sample aliquots from people with pulmonary tuberculosis, according to IUATLD classification [5,6].

GMNP (mg/mL)	Mean of AFB in NCBA (Number of Fields) *				SD			
	SSM-	SSM <9	SSM 1+	SSM 2+ and 3+	SSM-	SSM <9	SSM 1+	SSM 2+ and 3+
0.25	7 (65)	16 (193)	79 (422)	188 (306)	24	30	134	209
0.5	3 (39)	15 (187)	56 (424)	558 (310)	9	33	56	540
0.75	2 (32)	5 (128)	70 (306)	486 (310)	4	10	54	734
1.0	7 (47)	11 (163)	59 (391)	675 (310)	11	14	43	846
1.25	6 (102)	9 (294)	69 (293)	2030 (155)	12	15	49	2097
1.5	2 (33)	9 (155)	73 (244)	1740 (155)	3	16	46	1580
1.75	3 (31)	5 (94)	34 (141)	269 (149)	5	9	33	231
2.0	3 (19)	5 (99)	22 (105)	1206 (155)	11	10	41	1186
2.25	1 (15)	2 (36)	19 (48)	15 (45)	2	6	19	19
2.5	4 (89)	5 (184)	44 (191)	1891 (155)	11	10	41	1605
10.0	1 (76)	3 (136)	44 (445)	119 (278)	3	13	43	136

Abbreviations: SD: standard deviation. Classification based on IUATLD scale: Negative (0 AFB); Scanty (1–9 AFB); 1+ (10–99 AFB); 2+ (100–9,999 AFB); 3+ (>10,000 AFB). * Number of fields read in NCBA and SSM were from all independent samples (*n* = 23) and pools (*n* = 7).

**Table 5 biosensors-12-00029-t005:** Comparison of percentage differences in the mean number of quantified bacilli in smears evaluated by SSM and NCBA with GMNP (1.25 and 2.5 mg/mL) and with Tween 80 (5, 10, 15, and 20%) as a dispersant agent for GMNP-AFB complexes.

GMNP (mg/mL)	[Tween 80]	AFB Number (Mean)				No. Fields ^a^	Percentage Difference between TW with Respect NCBA and SSM			
		TW *	TW **	NCBA	SSM		TW *-NCBA	TW *-SSM	TW **-NCBA	TW **-SSM
1.25	5%	2400	956	804	441	3200	199	445	19	117
	10%	1915	914				138	335	14	107
	15%	1310	991				63	197	23	125
	20%	1335	825				41	158	3	87
2.5	5%	937	694	532			76	113	30	58
	10%	936	568				76	112	7	29
	15%	937	601				76	113	13	36
	20%	727	662				37	65	24	50

Abbreviation: TW: analysis after incubation with Tween 80 for 3 min (*) and 10 min (**). The letter ^a^ means: Sum of the 100 fields evaluated per smear obtained from aliquots of each pool. *n* = 4 pools.

## Data Availability

All data are contained within the article.

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
