# Peer review of "Tween 80 Improves the Acid-Fast Bacilli Quantification in the Magnetic Nanoparticle-Based Colorimetric Biosensing Assay (NCBA)"

_biosensors, 2022, doi:10.3390/bios12010029_

Round 1
Reviewer 1 Report
Review: Tween 80 improves the Acid-Fast Bacilli quantification in the magnetic nanoparticle-based colorimetric biosensing assay (NCBA).
In this study the authors investigate the effect of Tween 80 in the magnetic nanoparticle-based colorimetric biosensing assay (NCBA) when glycan-functionalized magnetic nanoparticles (GMNP) are used to detect acid-fast bacilli (AFB) related to tuberculosis.
They have published previously how the NCBA using this kind of nanoparticles improves the detection of AFB compared to conventional sputum smear microcroscopy (SSM). In the present study, the results show that when tween 80 is used in combination with GMNP at low concentrations, the results obtained by the NCBA are improved.
The main handicap of this paper is that the authors do not reproduced the previous results obtained in 2018 when high concentration of GMNP are used. The authors discuss what factors can affect to these differences and this is really appreciable and honest. Point e) states the influence of the batch of magnetic nanoparticles tested in this study as it is different to the one used in the previous study. In this context, the paper could be improved by adding information about the nanoparticles and other aspects.
The specific comments are listed below:
- Add a detailed experimental section about the synthesis of the nanoparticles (reactives, quantities, procedures, purification...). A reference is included, but the authors say that this procedure has been modified.
- Add details about the characteristics of the particles supported by characterization techniques (transmission electron microscopy, Dynamic light scattering, thermogravimetric analysis, etc). Note that not all the experiment should be included, but at least some of them that proved the correct synthesis and characterization.
- Tween 80 is a common surfactant used to improve the dispersability of nanoparticles and to improve the performance of multiple biosensing assays. Therefore, a depper discussion about the state of the art of this topic, including key references can improve the quality of the discussions.
- In line 338, the authors speak about results that have not been yet presented in the paper (use of GMNP at 10 mg/mL). A note sending the reader to the results presented later can help the redability of the manuscript.
- Table 4 shows AFB number while Figure 4 shows AFB means. The differences or relationships between these numbers are not clear and create some missunderstanding of the results.
Author Response
Reviewer 1. Review Report Form (Round 1)
General Comments. In this study the authors investigate the effect of Tween 80 in the magnetic nanoparticle-based colorimetric biosensing assay (NCBA) when glycan-functionalized magnetic nanoparticles (GMNP) are used to detect acid-fast bacilli (AFB) related to tuberculosis.
They have published previously how the NCBA using this kind of nanoparticles improves the detection of AFB compared to conventional sputum smear microscopy (SSM). In the present study, the results show that when tween 80 is used in combination with GMNP at low concentrations, the results obtained by the NCBA are improved.
The main handicap of this paper is that the authors do not reproduce the previous results obtained in 2018 when high concentration of GMNP are used. The authors discuss what factors can affect to these differences and this is really appreciable and honest. Point e) states the influence of the batch of magnetic nanoparticles tested in this study as it is different to the one used in the previous study. In this context, the paper could be improved by adding information about the nanoparticles and other aspects.
Response: We acknowledge reviewer’s comments and suggestions. In this study, we also analyzed 10 mg/mL of GMNP, but we did not perform more experiments due to inconsistency from our previous work. We discussed this discrepancy in Discussion section, lines 524-537, page 14. To improve our article, we address your observations in the following paragraphs.
Specific Comment 1. Add a detailed experimental section about the synthesis of the nanoparticles (reactives, quantities, procedures, purification...). A reference is included, but the authors say that this procedure has been modified.
Response: We have modified the Materials and Methods section to include a detailed description of the synthesis of the magnetic nanoparticles as follows:
See lines 187-195, page 4.
The core of GMNP was synthesized based on the procedure of Setterington et al. [44] by dissolving ferric chloride hexahydrate (1.08 g), sodium acetate (2.0 g) in ethylene glycol (30 mL) for 2 h at room temperature. The solution was transferred to a Teflon-lined stainless-steel pressure vessel (Parr Instrument Company), sealed, and heated at 200°C for 15 h. The pressure vessel was then cooled to room temperature and the synthesized nanoparticles were magnetically separated with a commercial magnetic separator (Promega Corporation, Madison, WI), washed with 20 mL of water three times and with 20 mL of ethanol three times, and dried overnight under vacuum. Chitosan was used to biofunctionalize the magnetic nanoparticle shell during synthesis.
Specific Comment 2. Add details about the characteristics of the particles supported by characterization techniques (transmission electron microscopy, Dynamic light scattering, thermogravimetric analysis, etc.). Note that not all the experiment should be included, but at least some of them that proved the correct synthesis and characterization.
Response: we have included the analyses that were used for the GMNP characterization in subsection Chemicals and reagents, to read as follows:
See lines 195-196, page 5.
A transmission electron microscope (TEM, JEOL 100CX) was used to visualize the GMNPs.
Specific Comment 3. Tween 80 is a common surfactant used to improve the dispersability of nanoparticles and to improve the performance of multiple biosensing assays. Therefore, a deeper discussion about the state of the art of this topic, including key references can improve the quality of the discussions.
Response: We have included the following paragraphs in the introduction and discussion, which include information on the dispersion of nanoparticles with Tween 80:
In the Introduction section,
See lines 150-158, page 4.
Polyoxyethylene sorbitan monooleate (commonly known as Tween 80), a nonionic surfactant, has been widely used as a reductant, dispersant, and stabilizer in the synthesis of metal nanoparticles, including those of gold [31], zinc [32], silver [33,34], and iron [35]. Tween 80 is unique in its ability to enhance steric repulsion between nanoparticles [31], demonstrating that when this surfactant is removed, nanoparticles aggregate, mainly when they are added in a solvent [36] under conditions of high ionic strength [37], as is the case in biological samples from the respiratory tract [38]. This aggregation reduces the functional surface area of nanoparticles, their mobility within the analysis matrix and their reactivity [39].
In the Discussion section,
See lines 426-434, page 12.
Surfactants such as Tween 80, are a type of organic compounds with polar (hydrophilic) and non-polar (hydrophobic) groups that disintegrate nanoparticles through the repulsive forces of static electricity, steric hindrance and Van der Waals force by their surface adsorption [49,50]. By changing the electric potential, surfactants increase the surface charge and repulsion between particles to reduce agglomeration [50]. The dispersant properties of Tween 80 have also been used in the disaggregation of mycobacteria in culture media [41] and it has been shown that it can affect bacterial adhesion by inhibiting the formation of biofilms, as occurs in Pseudomonas aeruginosa [51], Salmonella enterica [52] and Staphylococcus aureus [53], among others.
See lines 447-458, page 12.
In a study by Phenrat el al. [54], they evaluated the effect of iron nanoparticle concentration, size distribution (polydispersity) and magnetic attraction forces on agglomeration and transport in water-saturated sand column, determining that low concentrations of nanoparticles (0.03 mg/mL) could easily move in the sand column regardless of size or magnetic attraction forces between nanoparticles. Although we used different analytical conditions (nanoparticle design, analytical samples) and in our study we exceeded the "low" concentration evaluated by Phenrat el al. [54], we obtained similar results regarding the efficient use of low GMNP concentrations with respect to those that are higher. In this condition (low GMNP concentration and facilitated mobility), the diversity of Mtb cell wall receptors can form complexes with the GMNP by Brownian motion, hydrodynamic force, and physicochemical and molecular interactions [22].
See lines 471-474, page 13.
The increased visualization and quantification of AFB is facilitated by: (i) low GMNP concentration does not saturate the sample matrix as explained above; and (ii) the ability of Tween 80 to increase the separation between GMNP [31] and change mycobacterial cell wall morphology [55].
See lines 494-508, page 13
On the other hand, the evaluation of high GMNP concentrations (≥1.75 mg/mL) continued causing a greater number of GMNP-AFB complexes clusters, which reduced the visualization and quantification of AFB (Figure 2; Tables 2 and 4), as well as the proportion of positive fields enhanced with NCBA compared to SSM (Table 3). This effect was also observed by Phenrat et al. [54], where high concentrations of nanoparticles (up to 6 mg/mL) substantially increased agglomeration, due to binding between primary particles as between aggregates of them, a process that also becomes more sensitive to particle size distribution. In this sense, the addition of Tween 80 (5-20%) and incubation times did not improve the visualization and quantification of AFB. This possibly occurred due to significant Van der Waals interactions that increased the closer nanoparticles were to each other. At the same time, Brownian motion provided persistent collisions between nanoparticles leading to agglomeration [34], as well as to the fact that the observation of AFB depends on the saturation of receptors offered by Mtb cell wall glycolipids or glycoproteins to interact with GMNP [20] and on the formation of GMNP-AFB complexes per field observed under the microscope [21].
Specific Comment 4. In line 338, the authors speak about results that have not been yet presented in the paper (use of GMNP at 10 mg/mL). A note sending the reader to the results presented later can help the readability of the manuscript.
Response: We have added a note to redirect the reader to section 3.3, line 350, page 9.
Using GMNP at 10 mg/mL there was not improvement in the quantification of AFB, even after attempting to disaggregate the GMNP-AFB complexes with high concentrations of Tween 80 (10 and 20%) (see 3.3 section).
Specific Comment 5. Table 4 shows AFB number while Figure 4 shows AFB means. The differences or relationships between these numbers are not clear and create some misunderstanding of the results.
Response: Lines 408-411, page 11. In Table 5 (in the previous version Table 4), instead of the number of AFB we have included the mean numbers of AFB quantified.

Reviewer 2 Report
The authors based their study on the premise that even though it exhibits reduced sensitivity, sputum smear microscopy (SSM) remains a diagnostic test to detect the pathogen causing tuberculosis. As an alternative, they propose a nanoparticle-based colorimetric biosensing assay (NCBA). In addition, to optimize it, they evaluated the effect of the concentration of glucan-functionalized magnetic nanoparticles (GMNP) and Tween 80 on the detection of acid-fast bacilli through a comparative analysis of 225 sputum smears. Their report presumes that the smears analyzed by NCBA with GMNP concentrations ≤1.5 mg/mL showed better results than those analyzed by SSM. Furthermore, the detection significantly increased when included 5% Tween 80 for three minutes. Finally, they conclude that NCBA in these conditions can be useful to diagnose cases of paucibacillary in regions with limited resources.
Although their findings contribute to the tuberculosis diagnosis field, the following concerns must be addressed before publishing the study.
- Although the introduction shows a sufficient scientific basis, it is advisable to use a succinct writing style, as it is wordy on several points.
- The research methodology is standard; however, the scientific rigor is poor. Therefore, it is recommended to review the experimental approach and perform the research following a classic design of experiments protocol.
- Also, an extensive review of the study question or hypothesis is advisable, with rigorous adherence to the scientific method.
- Finally, a thorough English language review is recommended, preferably by a proofreading service that employs native speakers with scientific training.
Given the above, publication of the current manuscript version is not recommended.
Author Response
Reviewer 2. Review Report Form (Round 1)
General Comment. The authors based their study on the premise that even though it exhibits reduced sensitivity, sputum smear microscopy (SSM) remains a diagnostic test to detect the pathogen causing tuberculosis. As an alternative, they propose a nanoparticle-based colorimetric biosensing assay (NCBA). In addition, to optimize it, they evaluated the effect of the concentration of glucan-functionalized magnetic nanoparticles (GMNP) and Tween 80 on the detection of acid-fast bacilli through a comparative analysis of 225 sputum smears. Their report presumes that the smears analyzed by NCBA with GMNP concentrations ≤1.5 mg/mL showed better results than those analyzed by SSM. Furthermore, the detection significantly increased when included 5% Tween 80 for three minutes. Finally, they conclude that NCBA in these conditions can be useful to diagnose cases of paucibacillary in regions with limited resources.
Although their findings contribute to the tuberculosis diagnosis field, the following concerns must be addressed before publishing the study.
Specific Comment 1. Although the introduction shows a sufficient scientific basis, it is advisable to use a succinct writing style, as it is wordy on several points.
Response: In this new version, the authors have revised the reviewer's comments and, we hope that we have improved the writing style.
Specific Comment 2. The research methodology is standard; however, the scientific rigor is poor. Therefore, it is recommended to review the experimental approach and perform the research following a classic design of experiments protocol.
Response: We appreciate the reviewer’s observation, however, we believe that our current study has followed a full factorial experimental design with repeated measures and with a variable number of values, the latter mainly due to the intention of analyzing various concentrations of GMNP, Tween 80 and incubation times based on the homogeneity and volume of sputum samples. Natural clinical sputum samples, obtained from independent individuals, were used to evaluate the effect of addition of GMNP at different concentrations (NCBA assay) and compared to sputum smear microscopy (SSM). The NCBA assay, in a classical experiment design, is considered as an intervention or treatment group, whereas the SSM is considered the control group (with no intervention). We begin by evaluating different GMNP concentration to find the optimal concentration allowing us to detect and concentrate Mtb cells from natural clinical sputum samples (section 2.3.2). Then, to further improve the NCBA method, we tested the optimal GMNP concentrations obtained in the previous set of experiments (section 2.3.2) in combination with different Tween 80 concentrations (section 2.3.3).
These experiments led us to obtain results that the NCBA assay improves capture efficiency and increases AFB detection up to 445% over conventional SSM.
In addition, it is relevant to mention that, despite the differences in homogeneity and volume sputum samples, the inclusion in this study of "natural" biological sputum samples gives robustness to the reported findings. Other studies had been performed with samples inoculated with Mtb or BCG from healthy individuals (Qin et al. 2007; Ekrami et al. 2011) or with artificial sputum (Butler et al. 2020). This kind of samples are more homogeneous and do not have the microbiological complexity of other ones obtained from diseased persons neither the problem of the variability in the assays, thus there is a risk of optimizing the results in the analysis of such samples, which, ultimately, may well limit the applicability and the results of the method when used in real samples, which would lead to further optimization steps in the use of GMNP in NCBA.
Butler TE, Lee AJ, Yang Y, Newton MD, Kargupta R, Puttaswamy S, Sengupta S. 2020. Direct-from-sputum rapid phenotypic drug susceptibility test for mycobacteria. PLoS One. 15(8):1–21. doi:10.1371/journal.pone.0238298. http://dx.doi.org/10.1371/journal.pone.0238298.
Ekrami A, Samarbaf-Zadeh AR, Khosravi A, Zargar B, Alavi M, Amin M, Kiasat A. 2011. Validity of bioconjugated silica nanoparticles in comparison with direct smear, culture, and polymerase chain reaction for detection of Mycobacterium tuberculosis in sputum specimens. Int J Nanomedicine. 6:2729–2735. doi:10.2147/ijn.s23239.
Qin D, He X, Wang K, Zhao XJ, Tan W, Chen J. 2007. Fluorescent Nanoparticle-Based Indirect Immunofluorescence Microscopy for Detection of Mycobacterium tuberculosis. J Biomed Biotechnol. 2007:1–9. doi:10.1155/2007/89364. http://www.hindawi.com/journals/bmri/2007/089364/abs/.
Specific Comment 3. Also, an extensive review of the study question or hypothesis is advisable, with rigorous adherence to the scientific method.
Response: We have rewritten our research hypothesis within the main text as follows:
Lines 171-178, page 4.
Considering Tween 80 properties and our previous study observations [20], we hypothesize that combining of GMNP and Tween 80 in the NCBA method will improve the visualization and quantification of AFB from human clinical samples. To prove this hypothesis, we began by optimizing the GMNP concentration in clinical sputum samples. Then, we tested the optimal GMNP concentration in different Tween 80 concentrations. Therefore, the objective of this study was to evaluate GMNP concentration in NCBA in combination with Tween 80 to improve the quantification of AFB and increase the positivity of the observed fields.
Specific Comment 4. Finally, a thorough English language review is recommended, preferably by a proofreading service that employs native speakers with scientific training.
Response: We have performed a thorough English revision of the manuscript, however, due to limited time, if the reviewer and the journal editors consider that the manuscript should be reviewed by the journal's proofreading service, the authors agree, and we will support your decision
General Comment. Given the above, publication of the current manuscript version is not recommended.
Response: We appreciate your comment, we hope this new version clarifies your doubts and your requirements. The authors thank the reviewer for his time and contributions to improve the article.

Reviewer 3 Report
The manuscript describes Tween 80 can improves the accuracy of AFB with NCBA in SSM. And authors clamed that the AFB detection up to 445% over conventional SSM. This study may be useful to diagnose paucibacillary cases efficiently, rapidly and safely in regions with limited resources. The content of the article is good and interesting, but needs some revisions before publication. Required revisions are detailed below:
- Line 50, please indicate the country or region of the data.
- Line 57, the specific meaning of (1+, 2+, 3+) in this study should be explain.
- Replaced the references [7] with the latest data.
- Please standardize the writing of manuscripts, such as: Line116, (CFU/) mL.
- In the introduction, the principle of GMNP for SSM should be explained in more detail.
- Line 242, Is the room temperature of 18℃?
- In 3.3, why the incubation time of Tween 80 are 3 and 10 minutes and the concentration of GMNP are 1.5 and 2.5 mg/mL? What is the basis of this design?
- The principle of Tween 80 improving AFB quantification in NCBA should be analyzed.It would be helpful if authors could insert this content in the main manuscript.
Author Response
Reviewer 3. Review Report Form (Round 1)
General Comment. The manuscript describes Tween 80 can improves the accuracy of AFB with NCBA in SSM. And authors claimed that the AFB detection up to 445% over conventional SSM. This study may be useful to diagnose paucibacillary cases efficiently, rapidly and safely in regions with limited resources. The content of the article is good and interesting but needs some revisions before publication.
Specific Comment 1. Line 50, please indicate the country or region of the data.
Response: We have included the region of the data cited in the line 54, page 2.
In 2020, approximately 9.9 million people worldwide became ill from TB (of which only 5.8 were diagnosed and reported) and 1.5 million died, including 214,000 cases with TB and Human Immunodeficiency Virus (HIV) coinfection (WHO 2021).
Specific Comment 2. Line 57, the specific meaning of (1+, 2+, 3+) in this study should be explain.
Response: We have included a new Table (Table 1 on the new manuscript) and the following paragraph in the Introduction section to explain this categorization.
See lines 69-75, page 2.
According to the current guidelines for diagnosis by sputum smear microscopy (SSM) (Table 1) [5–7], of all notified new TB cases, 45% were diagnosed at 1+, 25% at 2+ and 27% at 3+ evidencing the diagnostic delay [8].
Table 1. AFB quantification guide and estimated AFB concentration per milliliter.
|
No. AFB Observed |
AFB Quantification |
Estimated AFB Concentration/mL |
|
0 in 100 or more fields |
Negative |
0 |
|
1-9 AFB/100 fields |
Scanty |
30,000 (3 x 104) |
|
10-99 AFB/100 fields |
1+ |
50,000 (5 x 104) |
|
1-10 AFB/field in at least 50 fields |
2+ |
100,000 (1 x 105) |
|
>10 AFB/field in at least 20 fields |
3+ |
500,000 (5 x 105) |
Source: International Union Against Tuberculosis and Lung Disease (IUATLD)/World Health Organization scale (WHO) [5–7].
Specific Comment 3. Replaced the references [7] with the latest data.
Response: We included the latest data by WHO Tuberculosis: Key facts October 14, 2021. Available online: https://www.who.int/news-room/fact-sheets/detail/tuberculosis (accessed on Nov 11, 2021)
Specific Comment 4. Please standardize the writing of manuscripts, such as: Line116, (CFU/) mL.
Response: We have standardized the terms in the manuscript:
Line 112, page 3.
Additionally, the authors determined that the NCBA assay could detect as low as 102 colony forming units per milliliter (CFU/mL).
Specific Comment 5. In the introduction, the principle of GMNP for SSM should be explained in more detail.
Response: We have described the principle of the use of GMNP in NCBA with respect to SSM.
Lines 120-128, page 3.
The core principle of NCBA is cell enrichment through GMNP, due to its glycan interaction with the Mtb cell envelope, which is rich in complex carbohydrates, glycoproteins and lipids [23,24]. These characteristics allow the formation of complexes between GMNP-AFB by Brownian motion, hydrodynamic force, and physicochemical and molecular interactions. Extraction of GMNP-AFB complexes is achieved using a simple magnet where GMNP acquire superparamagnetic properties, allowing the complexes to move in the direction of the electromagnetic field and generate a highly enriched crowding effect [22]. The increase in AFB number favored by GMNP makes a substantial difference compared to unconcentrated SSM.
Specific Comment 6. Line 242, Is the room temperature of 18℃?
Response: Yes, we mean the room temperature. At San Cristóbal de Las Casas, Chiapas, Mexico (study site), at the time the study was conducted, the average temperature was usually 18 degrees Celsius
Specific Comment 7. In 3.3, why the incubation time of Tween 80 are 3 and 10 minutes and the concentration of GMNP are 1.5 and 2.5 mg/mL? What is the basis of this design?
Response: We addressed these questions as follows:
About incubation times with Tween 80. Usually, Tween 80 is added to Mtb cultures at very low concentrations (0.05%) and left to incubate (along with culture additives) for weeks (Stoops et al. 2010; Wang et al. 2011; Pietersen et al. 2019). In our study, to speed up this process, we carried out several proofs evaluating high concentrations of Tween 80 (5-20%) and short incubation times. It is known that the addition of high Tween 80 concentrations, mycobacterial cell wall components are loss rendering change in its conformation (Van Boxtel et al. 1990; Masaki et al. 1990; Leisching et al. 2016). This conformational change aids in the release of bacilli from the complex with GMNP.
On the other hand, since high exposure times could affect the mycobacteria, we analyzed short incubation times to avoid overexposure of the aliquot of the samples to the homogenizing solution and Tween 80 (before making the smear and ZN staining). Therefore, the established incubation time was experimental determined so that Tween 80 would have a favorable effect on the modification of the Mtb envelope to disintegrate the GMNP-AFB complexes in a short time.
Leisching G, Pietersen R-D, Wiid I, Baker B. 2016. Virulence, biochemistry, morphology and host-interacting properties of detergent-free cultured mycobacteria: An update. Tuberculosis. 100:53–60. doi:10.1016/j.tube.2016.07.002. https://linkinghub.elsevier.com/retrieve/pii/S1472979216301354.
Masaki S, Sugimori G, Okamoto A, Imose J, Hayashi Y. 1990. Effect of Tween 80 on the Growth of Mycobacterium avium Complex. Microbiol Immunol. 34(8):653–663. doi:10.1111/j.1348-0421.1990.tb01041.x. https://onlinelibrary.wiley.com/doi/10.1111/j.1348-0421.1990.tb01041.x.
Stoops JK, Arora R, Armitage L, Wanger A, Song L, Blackburn MR, Krueger GR, Risin SA. 2010. Certain surfactants show promise in the therapy of pulmonary tuberculosis. In Vivo (Brooklyn). 24(5):687–694.
Wang C, Mahrous EA, Lee RE, Vestling MM, Takayama K. 2011. Novel Polyoxyethylene-Containing Glycolipids Are Synthesized in Corynebacterium matruchotii and Mycobacterium smegmatis Cultured in the Presence of Tween 80. J Lipids. 2011:1–12. doi:10.1155/2011/676535. http://www.hindawi.com/journals/jl/2011/676535/.
Pietersen RD, du Preez I, Loots DT, van Reenen M, Beukes D, Leisching G, Baker B. 2019. Tween 80 induces a carbon flux rerouting in Mycobacterium tuberculosis. J Microbiol Methods. 170(105795):1–16. doi:10.1016/j.mimet.2019.105795. https://doi.org/10.1016/j.mimet.2019.105795.
Van Boxtel RM, Lambrecht RS, Collins MT. 1990. Effects of colonial morphology and Tween 80 on antimicrobial susceptibility of Mycobacterium paratuberculosis. APMIS. 98:901–908. doi:10.1128/AAC.34.12.2300.
Regarding GMNP concentrations. Upon identifying that low GMNP concentrations (0.25, 1.0 and 1.25 mg/mL) improved AFB quantification and high concentrations (≥ 1.75 mg/mL) reduced it, the authors decided to evaluate a low and high concentration of GMNP to determine the dispersing effect of Tween 80 under the two conditions. Analyzed low concentrations of GMNP improved the classification of negative SSM fields compared to the rest of the concentrations evaluated (Table 2), but in additional exploratory analysis we identified that the concentration of 1.25 significantly improved the average number of AFB compared to the other concentrations (with t-student test, p<0.05). In this sense, we decided to only include the 2.5 mg/mL concentration as a high concentration, because the 10 mg/mL concentration obtained the worst percentage of improvement with respect to negative SSM fields (Table 2).
We included these clarifications in the paper:
Lines 265-273, page 6-7.
- b) Due to the limited results obtained in the previous procedure, we further evaluated GMNP concentrations (low-high) and Tween 80 at different concentrations (5, 10, 15 and 20% v/v) and two incubation times (3 and 10 min) experimentally determined by the authors to favor the modification of the mycobacterial cell wall, without exceeding the interaction time of the aliquots with the homogenizing solution and Tween 80. We analyzed 32 aliquots of approximately 1 mL obtained from 4 pools: 16 were concentrated with GMNP at 1.5 (the optimal low concentration analyzed, p<0.05) and 16 with 2.5 mg/mL (the highest concentration analyzed, before the 10 m/mL which showed the worst results).
Specific Comment 8. The principle of Tween 80 improving AFB quantification in NCBA should be analyzed. It would be helpful if authors could insert this content in the main manuscript.
Response: We have modified the following paragraph:
Lines 159-170, page 4.
In mycobacterial culture media, Tween 80 is added as an alternative energy source for growth [40] and to obtain homogeneous Mtb suspensions by reducing cell clumping [41]. It has been suggested that the interaction of Tween 80 with mycobacterial cell wall components such as phthiocerol dimycocerosate (PDIM) [40], arabinomannan (AM), trehalose dimycolate (TDM) and mycolic acids induces solubilization and permeabilization, altering the morphology of the bacilli [41–43]. Thus, we theorize that AFB quantification could be improved by adding Tween 80 in NCBA due to its effects of dispersing clumped GMNP more stably in aqueous solutions with the presence of biological molecules (e.g. nucleic acids or proteins of different sizes, among others) [31] and the conformational changes in the Mtb cell envelope induced by Tween 80 could favor the disassembly of GMNP-AFB complexes, which would eventually allow their release as single bacilli without GMNP coating and improve AFB quantification in NCBA.

Round 2
Reviewer 1 Report
The authors have addressed all the comments and the article has been notably improved. So it is almost ready to be published. I have just a couple of coments about the synthesis os NPs:
- The synthesis of Nanoparticles is correctly addressed in lines 187-195, page 4. . However, their functionalization is also a key step and it is not described. Please, describe it appropriately.
- A transmission electron microscope (TEM, JEOL 100CX) was used to visualize the GMNPs, but they do not specify how they prepare the sample and how they calculate the diameter of the NPs (software?), neither any image of the NPs is shown. Please, address this.
Author Response
Reviewer 1. Review Report Form Round 2
Comments and Suggestions for Authors
General Comment. The authors have addressed all the comments and the article has been notably improved. So, it is almost ready to be published. I have just a couple of comments about the synthesis of NPs:
Response: Thank you very much for all your comments, observations, and contributions, all have been enriching and were helpful in improving our manuscript. We will address your specific comments below.
Specific Comment 1. The synthesis of Nanoparticles is correctly addressed in lines 187-195, page 4. However, their functionalization is also a key step, and it is not described. Please, describe it appropriately.
Response: Thank you for your comment. We agree with you. We have included the following paragraph in the Materials and methods section in Lines 187-197, pages 4-5.
The GMNP was synthesized based on the procedure of Setterington et al. [44] with modification, by dissolving ferric chloride hexahydrate (1.08 g), sodium acetate (2.0 g) in ethylene glycol (30 mL) for 2 h at room temperature. The solution was transferred to a Teflon-lined stainless-steel pressure vessel (Parr Instrument Company), sealed, and heated at 200°C for 15 h. The pressure vessel was then cooled to room temperature. The synthesized nanoparticles were magnetically separated with a commercial magnetic separator (Promega Corporation, Madison, WI), washed with 20 mL of water three times and with 20 mL of ethanol three times, and dried overnight under vacuum. Chitosan was used to biofunctionalize the magnetic nanoparticles during synthesis by adding chitosan in the ferric chloride-sodium acetate-ethylene glycol solution, essentially coating the shell of the nanoparticles with D-glucoseamine or glycosaminoglycan units.
Specific Comment 2. A transmission electron microscope (TEM, JEOL 100CX) was used to visualize the GMNPs, but they do not specify how they prepare the sample and how they calculate the diameter of the NPs (software?), neither any image of the NPs is shown. Please, address this.
Response: We have included the procedures for preparing the sample for TEM and how the nanoparticle diameter measurement was performed in Lines 198-215, page 5.
To visualize the GMNPs, a transmission electron microscope (TEM, JEOL 100CX) from the Center for Advance Microscopy, Michigan State University was used. A TEM image of GMNPs is shown in Figure 1 (adopted from Bhusal et al. [21]). The image was prepared using the negative staining method [45]. Briefly, the procedure used was the droplet technique. A drop of the magnetic nanoparticle solution was added on a copper electron microscope (EM) grating surface pretreated with 300-400 mesh. The EM grid with adsorbed particles was washed with deionized water to remove salts and macromolecules that could interfere with particle staining. Next, the EM grid was stained (e.g., with 1% uranyl acetate) to produce a thin amorphous film after drying with filter paper to reveal the final structural details of the particles. Finally, the EM grating was placed under the TEM (at 20,000x magnification) for visualization and image capture. The size of the GMNPs (non-clustered nanoparticles) was measured using the Keyence VK-X150 laser scanning confocal microscope system measurement software (Nano-Biosensors Lab, Michigan State University).
[Figure attached in word version]
Figure 1. Transmission electron microscope (TEM) image of the glycan-coated magnetic nanoparticle clusters, with several iron oxides enclosed in the glycan polymer. Some nanoparticles are protruding from the cluster. Adopted from Bhusal et al. [21].

Reviewer 2 Report
The authors adequately addressed my suggestions, recommendations, and concerns. For this reason, I recommend publication of the manuscript after a thorough final review.
Author Response
Reviewer 2. Review Report Form Round 2
Comments and Suggestions for Authors
General Comment. The authors adequately addressed my suggestions, recommendations, and concerns. For this reason, I recommend publication of the manuscript after a thorough final review.
Response: Thank you very much for this, and your previous comments, all of which have been enriching to our work.
